# Stability Analysis and Control Strategy Optimization of a Paralleled IPOS Phase-Shifted Full-Bridge Converters System Based on Droop Control

Zhenghao Qin [1], Huafeng Cai [1,*] and Xinchun Lin [2]

1  School of Electrical & Electronic Engineering, Hubei University of Technology, Wuhan 430068, China; 102100233@hbut.edu.cn
2  School of Electrical & Electronic Engineering, Huazhong University of Science and Technology, Wuhan 430074, China; linxinchun@hust.edu.cn
*  Correspondence: whgkzj@hbut.edu.cn

**Abstract:** The application of high-power DC equipment further increases the power supply scale of DC systems. But, it is difficult for a single converter to support high transmission power, so multiple converters must operate in parallel for efficient power transmission. In a parallel system comprising many IPOS phase-shifting full-bridge converters, current sharing can be realized via droop control. However, the stability of the parallel system using current-sharing control will appear poor in light load conditions, so it is necessary to analyze the stability of parallel systems in light load conditions. Firstly, a single IPOS phase-shifted full-bridge control system is modeled; on this basis, the state space model of the n-module paralleled IPOS phase-shifted full-bridge converters system is derived. Then, the influence of load power and the number of parallel IPOS phase-shifted full-bridge converters on the system stability is analyzed via eigenvalue analysis, and an optimal control strategy based on a particle swarm optimization algorithm is proposed. The control parameters are optimized for the parallel system of eight IPOS phase-shifted full-bridge converters. Finally, the above results are simulated to verify the accuracy of the stability analysis and the feasibility of the optimized control strategy.

**Keywords:** n-module paralleled IPOS phase-shifted full-bridge converters system; droop control; stability; eigenvalue analysis; control strategy

## 1. Introduction

With the continuous development of DC converters, phase-shifted full-bridge converters are widely used in medium- and high-power supply applications due to their simple structure and high efficiency, and have gradually become the mainstream topology. When the output side is required to be medium and high voltage, multiple phase-shifted full-bridge topologies can be connected via input parallel and output series (IPOS) to form an IPOS phase-shifted full-bridge converter to achieve a higher boost ratio. In recent years, with the development of new energy vehicles, aerospace, shipbuilding, and other fields, high-power DC power supply has been widely used [1]. At the same time, the application of high-power DC equipment has led to a further increase in the scale of the DC power supply system, and the power level that the DC converter needs to transmit is also growing higher and higher. A single converter is difficult to support higher transmission power, and multiple converter modules need to be used in parallel for power transmission [2].

A modular parallel power supply can greatly increase the redundancy, maintainability, and reliability of a system [3]. When multiple IPOS phase-shifted full-bridge converter modules constitute a parallel system, since the output characteristics of the converter using voltage stabilization control are approximately voltage sources, the problem of uneven output currents is prone to occur in the parallel system, which may lead to high thermal stress on the power device and even the breakdown of the power device [4]. Therefore,

current-sharing control technology is often used to achieve the average distribution of the output current between converters [5]. In [6–9], droop control was used to achieve current sharing in parallel systems. However, the parallel system using current-sharing control has poor stability and a significant decline in dynamic performance under light load conditions, which adversely affects the safety and stability of the DC power supply system. The existing engineering practice generally adopts the method of parallel dead load on the output side to avoid the system entering the working state of too small a load and ensure the reliability of parallel system operation. But the load will greatly increase power consumption and reduce system efficiency. Therefore, it is of great significance to analyze the stability of the parallel system under a light load and improve the system stability by optimizing the parameters.

For the existing research on the stability of the DC parallel system, in [10], the stability of the two-module paralleled buck converters is analyzed via a bifurcation diagram. In [11], the stability of the n input series output parallel (ISOP) DC–DC converter parallel system is analyzed based on the Routh criterion, and the stable range of controller parameters is obtained. An extended stability analysis method of a DC–DC converter parallel system considering periodic disturbances based on the Floquet theory is proposed in [12], which can be applied to the stability analysis of the multi-module parallel system. In [13], a state space model is established based on a parallel system composed of two buck converters using droop control. The influence of the droop coefficient and controller parameters on system stability is analyzed using characteristic roots. The authors of [14] study the stability of charging stations with a parallel structure of multi-controllable rectifier modules and the load capacity of a cascaded system of converters using the impedance analysis method. In [15], large-signal modeling is performed to such a system using the controlled current sources to represent the parallel-connected converters, and the mixed potential theory-based stability criterion is then generated with the simplified model. Finally, the influence of the key parameters on the system stability is studied according to the criterion. The existing literature does not study the stability of the parallel system with droop control under light load conditions and does not pay much attention to the problem of parallel dead load on the output side of the parallel system. In addition, when the power level of the DC converter to be transmitted is higher, the number of converters that the system needs to be connected in parallel will increase accordingly. Therefore, the change in system performance when the number of converters in parallel increases also needs to be further explored.

In [16], the theoretical stability analysis and control of a supercapacitor based on SOSM for an MEA application are presented to guarantee system stability. On this basis, in [17], a novel control design based on second-order sliding mode control and uniting control is proposed for the buck–boost converter, which overcame the difficulties generated via the nonlinear input gain function of the system not being sign definite. In addition, many algorithms have been used to optimize the control parameters to improve the stability and dynamic performance of the system. In [18,19], an adaptive velocity particle swarm optimization algorithm and a genetic simulated annealing algorithm are proposed to optimize the control parameters. The objects of these papers are all nonlinear systems. For nonlinear systems, after linearization via state space modeling, the system stability can be analyzed using the position of the characteristic root so that the characteristic root can be related to the objective function. Given the above problems, this paper combines the eigenvalue analysis method, takes the penalty function of the distance between the real part of the eigenvalue and the imaginary axis and the damping ratio as the objective function, and proposes a global optimization strategy of control parameters based on particle swarm optimization (PSO), which effectively improves the system stability.

This paper first establishes the mathematical model of an n-module paralleled IPOS phase-shifted full-bridge converters system based on droop control, focuses on the light load condition, analyzes the stability of the paralleled IPOS phase-shifted full-bridge converters system, and studies the influence of load power and the number of parallel converters on the system stability. Then, an optimal control strategy based on the PSO algorithm

is proposed to improve the stability of the eight-module paralleled IPOS phase-shifted full-bridge converters system under light load conditions, and the problem of parallel dead load on the output side is solved. Finally, the system model is built in MATLAB/Simulink, and the correctness of the theoretical analysis results is verified via simulation experiments.

## 2. Mathematical Model of the Paralleled IPOS Phase-Shifted Full-Bridge Converter System

The single-module IPOS phase-shifted full-bridge converter system is shown in Figure 1. It consists of two phase-shifted full-bridge modules (PSHB modules) with parallel input and series output (IPOS). This structure is suitable for low-voltage high-current applications on the input side and high-voltage low-current applications on the output side. In Figure 1, $Q_{11} \sim Q_{18}$ are the switching tubes, $C_{r1} \sim C_{r8}$ are the parasitic capacitance of the switching tubes, $D_1 \sim D_8$ are the rectifier diodes, $C_{b1}$ and $C_{b2}$ are the blocking capacitors, $R_{s1}C_{s1}D_{s1}$ and $R_{s2}C_{s2}D_{s2}$ are two RCD absorption circuits, $T_{r1}$ and $T_{r2}$ are transformers, 1:$K$ is the ratio of the primary and secondary sides of the transformer, $L_{lk1}$ and $L_{lk2}$ are transformer leakage inductances, $L_f$ is the output filter inductor, $i_L$ is the output filter inductor current, $C_f$ is the output filter capacitor, $u_c$ is the output filter capacitor voltage, $i_o$ is the output current, $R_o$ is the output resistance, $U_o$ is the output voltage, and $U_{in}$ is the input voltage.

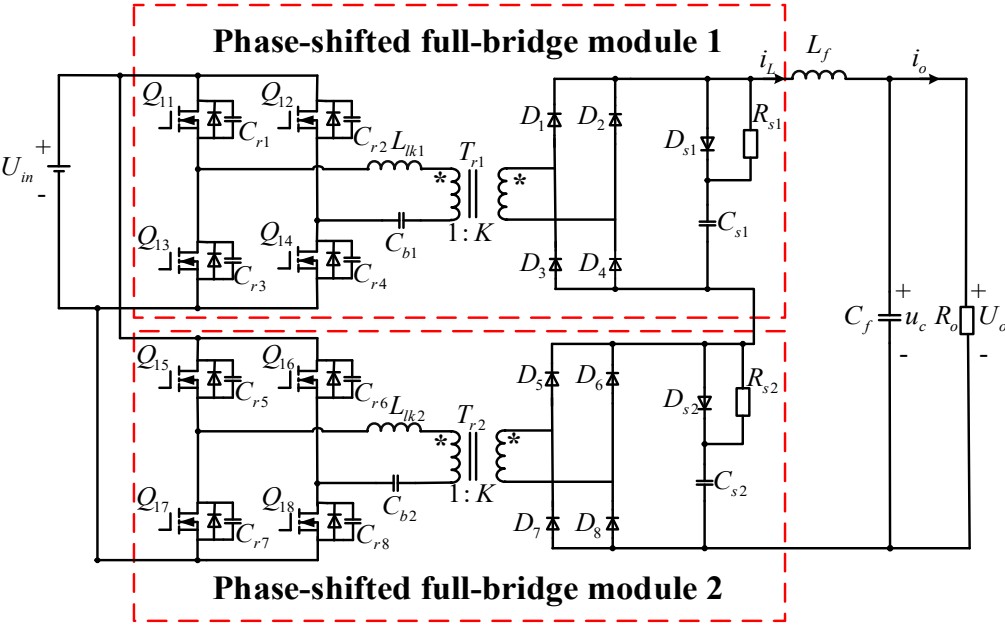

**Figure 1.** IPOS phase-shifted full-bridge converter circuit diagram.

In this section, the n-module paralleled IPOS phase-shifted full-bridge converters system is modeled, as shown in Figure 2. The system adopts droop control to realize the average distribution of the output current between n IPOS phase-shifted full-bridge converters. The control loop adopts a single voltage loop control. At the same time, considering the digital control delay, the mathematical model of the single-module IPOS phase-shifted full-bridge control system is first established. On this basis, the mathematical model of the n-module paralleled IPOS phase-shifted full-bridge converters system is established.

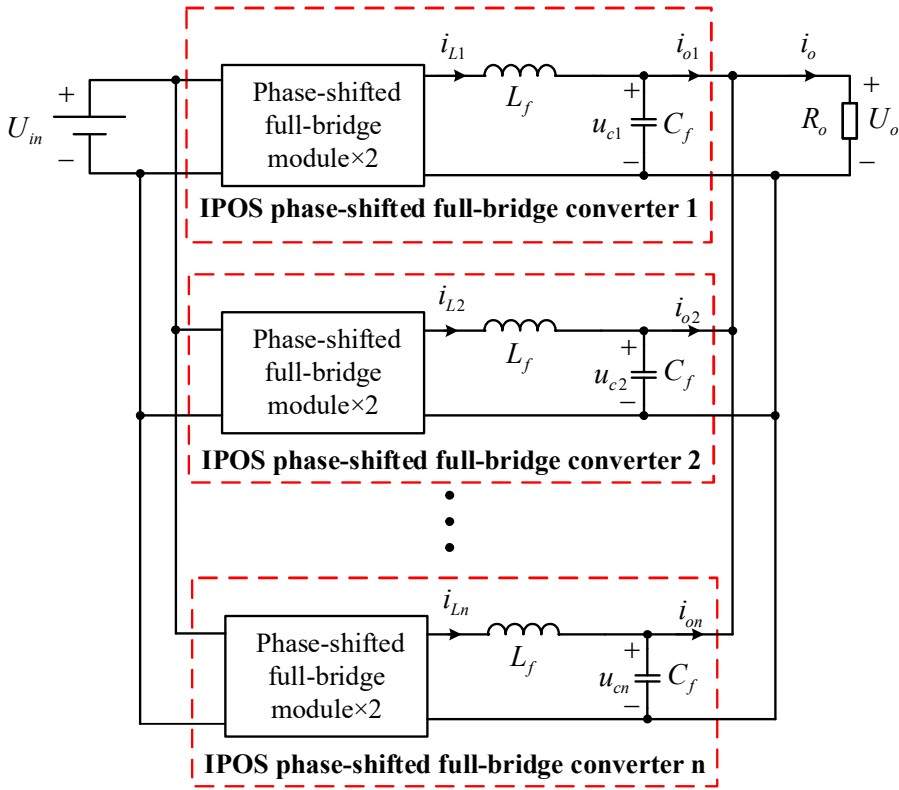

**Figure 2.** Paralleled system composed of n IPOS phase-shifted full-bridge converters.

### 2.1. Modeling of Single-Module IPOS Phase-Shifted Full-Bridge System

Firstly, the phase-shifted full-bridge converter is analyzed. On this basis, the small signal model of the IPOS phase-shifted full-bridge converter is constructed.

For the phase-shifted full-bridge converter, there is a problem with duty cycle loss during the commutation of the rectifier diode. It can be seen from [20,21] that the effective duty cycle of the secondary side of the transformer can be expressed as follows:

$$d_{eff} = d - \frac{4KL_{lk}I_Lf_s}{U_{in}} + \frac{4C_rU_{in}f_s}{KI_L} \tag{1}$$

where $f_s$ is the switching frequency, and $C_r$ is the parasitic capacitance of the switching tube. $d_{eff}$ is affected by the input voltage $U_{in}$, the primary duty cycle $d$, and the inductor current $I_L$, so there are three kinds of disturbances $\hat{d}_u$, $\hat{d}_u$, and $\hat{d}_i$ in the small signal model, which are expressed as follows:

$$\hat{d}_u = (\frac{4KL_{lk}I_Lf_s}{U_{in}{}^2} + \frac{4C_rf_s}{KI_L})\hat{u}_{in} = \frac{I_LR_d}{U_{in}{}^2}\hat{u}_{in} \tag{2}$$

$$\hat{d}_d = \hat{d} \tag{3}$$

$$\hat{d}_i = -(\frac{4KL_{lk}I_Lf_s}{U_{in}} + \frac{4C_rU_{in}f_s}{KI_L{}^2})\hat{i}_L = -\frac{R_d}{U_{in}}\hat{i}_L \tag{4}$$

In the above formulas, $\hat{u}_{in}$, $\hat{d}$, and $\hat{i}_L$ are the small disturbance of the input voltage, the small disturbance of the duty ratio of the primary side of the converter, and the small disturbance of the output filter inductor current, respectively. $R_d$ can be expressed as follows:

$$R_d = 4KL_{lk}f_s + \frac{4C_rR_o{}^2U_{in}{}^2f_s}{KU_c{}^2} \tag{5}$$

Since the two phase-shifted full-bridge modules in the IPOS phase-shifted full-bridge converter adopt the same control system, the control signals of the two phase-shifted full-bridge modules only have the phase difference of $\pi/2$. The relationship between the output voltage of the rectifier bridge of the IPOS phase-shifted full-bridge converter and the output voltage of the rectifier bridge of each module can be expressed as follows:

$$U_\Sigma = KU_{in}(d_{eff1} + d_{eff2}) = 2KU_{in}d_{eff} \tag{6}$$

Among them, $d_{eff1}$ and $d_{eff2}$ are the effective duty cycles of the two phase-shifted full-bridge modules, respectively. Further, the small disturbance of the output voltage of the rectifier bridge of the IPOS phase-shifted full-bridge converter is expressed as follows:

$$\hat{u}_\Sigma = 2K\hat{u}_{in}d_{eff} + 2KU_{in}(\hat{d}_u + \hat{d}_d + \hat{d}_i) = 2K(d_{eff} + \frac{I_L R_d}{U_{in}})\hat{u}_{in} - 2KR_d\hat{i}_L + 2KU_{in}\hat{d} \tag{7}$$

The small disturbance equations of rectifier bridge output voltage and filter inductor current are established as follows:

$$\hat{u}_\Sigma = L_f \frac{d\hat{i}_L}{dt} + \hat{u}_c \tag{8}$$

$$\hat{i}_L = C_f \frac{d\hat{u}_c}{dt} + \frac{\hat{u}_c}{R_o} \tag{9}$$

Combining (7)–(9) and ignoring the disturbance of the input voltage, we can obtain the following:

$$\hat{i}_L = \frac{2KU_{in}\hat{d} - \hat{u}_c}{sL_f + 2KR_d} \tag{10}$$

$$\hat{u}_c = \frac{\hat{i}_L - \hat{i}_o}{sC_f} \tag{11}$$

In the above formulas, $\hat{u}_c$ is the small disturbance of the output capacitor voltage. From (10) to (11), the small signal model of the IPOS phase-shifted full-bridge converter is further obtained as follows.

In Figure 3, $\hat{i}_o$ is the small disturbance of the output current. The control part adopts droop control, the control loop is a single voltage loop, and the controller is a proportional integral (PI) regulator.

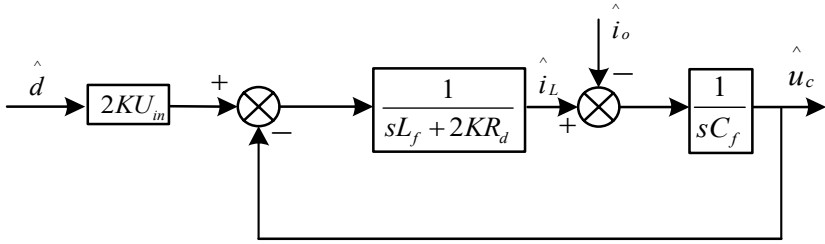

**Figure 3.** Small signal model of IPOS phase-shifted full-bridge converter.

Computational and PWM delays exist in the actual control system and, generally, the delays are considered $1.5T_s$ and modeled using $e^{-1.5T_s s}$. However, it is a nonlinear element and inconvenient for calculation and mathematical derivation. Thus, we chose to use the Padé approximation.

The Padé approximation of $e^{-\tau s}$ is expressed as follows [22]:

$$e^{-\tau s} = \frac{1 - \frac{\tau s}{2} + \frac{\tau^2 s^2}{12} + \cdots + (-1)\frac{n!\tau^n s^n}{(2n)^2}}{1 + \frac{\tau s}{2} + \frac{\tau^2 s^2}{12} + \cdots + \frac{n!\tau^n s^n}{(2n)^2}} \tag{12}$$

Therefore, the first-order Padé approximation for $e^{-1.5T_s s}$ is expressed as follows:

$$e^{-1.5T_s s} = \frac{1 - 0.75T_s s}{1 + 0.75T_s s} \tag{13}$$

To better select the state variables and simplify the subsequent modeling, the delay link is further decomposed into a proportional link and a first-order inertial link.

$$e^{-1.5T_s s} = \frac{1 - 0.75T_s s}{1 + 0.75T_s s} = -1 + \frac{2}{1 + 0.75T_s s} \tag{14}$$

Based on the above analysis, a small signal model of a single IPOS phase-shifted full-bridge control system can be obtained, as shown in Figure 4.

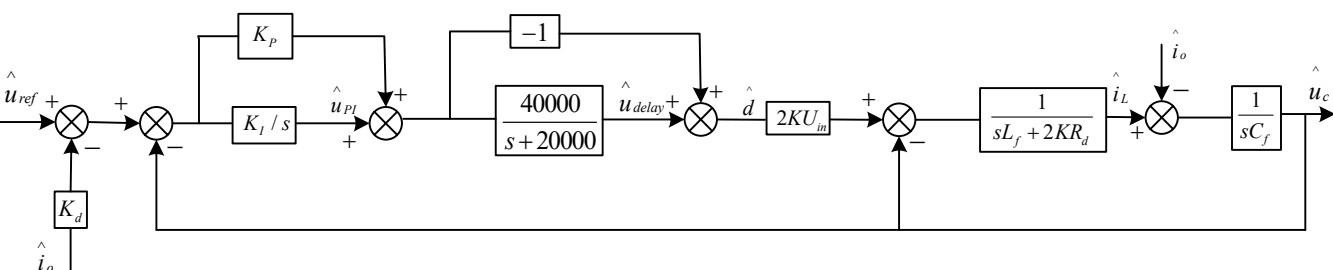

**Figure 4.** Small signal model of IPOS phase-shifted full-bridge control system.

In Figure 4, $\hat{u}_{ref}$ is the small disturbance of the reference voltage, $K_P$ and $K_I$ are the proportional coefficient and integral coefficient of the voltage loop, respectively, and $K_d$ is the droop coefficient of the droop control. To establish the state space model of a single-module IPOS phase-shifted full-bridge control system, the inductor current disturbance $\hat{i}_L$, the disturbance $\hat{u}_{PI}$ output via the converter voltage loop integrator, the disturbance $\hat{u}_d$ output by the delay part of the inertial link, and the disturbance $\hat{u}_c$ of the capacitor voltage are selected as the state variables, namely $x_1 = [\hat{i}_L \ \hat{u}_{PI} \ \hat{u}_d \ \hat{u}_c]^T$, and $\hat{u}_{ref}$ is selected as the input quantity, namely $u = [\hat{u}_{ref}]$. According to Figure 4, the relationship between the state variables can be obtained as follows:

$$\hat{i}_L = \frac{2KU_{in}(\hat{u}_d - \hat{u}_{PI} + K_d K_P \hat{i}_o + K_P \hat{u}_c - K_P \hat{u}_{ref}) - \hat{u}_c}{sL_f + 2KR_d} \tag{15}$$

$$\hat{u}_{PI} = \frac{K_I}{s}(\hat{u}_{ref} - K_d \hat{i}_o - \hat{u}_c) \tag{16}$$

$$\hat{u}_d = \frac{2}{1 + 0.75T_s s}(\hat{u}_{PI} - K_d K_P \hat{i}_o - K_P \hat{u}_c + K_p \hat{u}_{ref}) \tag{17}$$

$$\hat{u}_c = \frac{1}{sC_f}(\hat{i}_L - \hat{i}_o) \tag{18}$$

Furthermore, the state matrix $A_1$ and the input matrix $B_1$ in the state equation $\dot{x}_1 = A_1 x_1 + B_1 u$ of the single-module IPOS phase-shifted full-bridge control system can be expressed as follows:

$$A_1 = \begin{bmatrix} -\frac{2KR_d}{L_f} & -\frac{2KU_{in}}{L_f} & \frac{2KU_{in}}{L_f} & \frac{2KU_{in}K_p(K_d+R_o)-R_o}{R_o L_f} \\ 0 & 0 & 0 & -\frac{K_I K_d}{R_o} - K_I \\ 0 & \frac{2}{0.75T_s} & -\frac{1}{0.75T_s} & \frac{2(K_d+R_o)K_P}{0.75T_s R_o} \\ \frac{1}{C_f} & 0 & 0 & -\frac{1}{R_o C_f} \end{bmatrix} \tag{19}$$

$$B_1 = \begin{bmatrix} -\frac{2KU_{in}K_P}{L_f} & K_I & \frac{2K_P}{0.75T_s} & 0 \end{bmatrix}^T \tag{20}$$

### 2.2. Modeling of N-Module Paralleled IPOS Phase-Shifted Full-Bridge Converters System

The n-module paralleled IPOS phase-shifted full-bridge converters system is composed of n IPOS phase-shifted full-bridge converters with the same parameters, which are connected via input parallel and output parallel, as shown in Figure 2. When the output is in parallel, the capacitors of each converter are in parallel, and the capacitor voltage is equal to the output voltage, namely $u_{c1} = \cdots = u_{cn} = u_o$. Therefore, the capacitor voltage disturbance of each converter can be coupled into the voltage disturbance of a parallel equivalent capacitor, denoted as $\hat{u}_c'$. According to the single machine derivation, each converter can select three state variables $\hat{i}_L$, $\hat{u}_{PI}$, and $\hat{u}_d$ as the state variables of the whole parallel system model. Therefore, n parallel systems take the state variables as follows:

$$x_n = [\hat{i}_{L1} \ \hat{u}_{PI1} \ \hat{u}_{d1} \ \hat{i}_{L2} \ \hat{u}_{PI2} \ \hat{u}_{d2} \ \cdots \ \hat{i}_{Ln} \ \hat{u}_{PIn} \ \hat{u}_{dn} \ \hat{u}_c']^T \tag{21}$$

The state matrix $A_n$ in the state space model is a $3n + 1$ order.

The relationship between the state variables in the x-th converter can be derived from a single machine as follows:

$$
\begin{aligned}
s\hat{i}_{Lx} = \ & [\frac{2(n-1)KU_{in}K_dK_P}{nL_f} - \frac{2KR_d'}{L_f}]\hat{i}_{Lx} - \frac{2KU_{in}K_dK_P}{nL_f}(\hat{i}_{L1} + \cdots + \hat{i}_{L(x-1)} + \hat{i}_{L(x+1)} + \cdots + \hat{i}_{Ln}) \\
& - \frac{2KU_{in}}{L_f}\hat{u}_{PIx} + \frac{2KU_{in}}{L_f}\hat{u}_{dx} + (\frac{2KU_{in}K_P}{L_f} + \frac{2KU_{in}K_dK_P}{nR_oL_f} - \frac{1}{L_f})\hat{u}_c' - \frac{2KU_{in}K_P}{L_f}\hat{u}_{ref}
\end{aligned}
\tag{22}
$$

$$
\begin{aligned}
s\hat{u}_{PIx} = \ & -\frac{(n-1)K_dK_I}{n}\hat{i}_{Lx} + \frac{K_dK_I}{n}(\hat{i}_{L1} + \cdots + \hat{i}_{L(x-1)} \\
& + \hat{i}_{L(x+1)} + \cdots + \hat{i}_{Ln}) - (\frac{K_dK_I}{nR_o} + K_I)\hat{u}_c' + K_I\hat{u}_{ref}
\end{aligned}
\tag{23}
$$

$$
\begin{aligned}
s\hat{u}_{dx} = \ & -\frac{2(n-1)K_dK_P}{0.75nT_s}\hat{i}_{Lx} + \frac{2K_dK_P}{0.75nT_s}(\hat{i}_{L1} + \cdots + \hat{i}_{L(x-1)} + \hat{i}_{L(x+1)} + \cdots + \hat{i}_{Ln}) \\
& + \frac{2}{0.75T_s}\hat{u}_{PIx} - \frac{1}{0.75T_s}\hat{u}_{dx} - (\frac{2K_dK_P}{0.75nT_sR_o} + \frac{2K_P}{0.75T_s})\hat{u}_c' + \frac{2K_P}{0.75T_s}\hat{u}_{ref}
\end{aligned}
\tag{24}
$$

$$s\hat{u}_{cx} = \frac{1}{nC_f}(\hat{i}_{L1} + \cdots + \hat{i}_{Ln}) - \frac{1}{nC_fR_o}\hat{u}_c' \tag{25}$$

Among them, $R_d' = 4KL_{lk}f_s + \frac{4n^2C_rR_o^2U_{in}^2f_s}{KU_c^2}$. For the relationship between the three state variables $\hat{i}_L$, $\hat{u}_{PI}$, and $\hat{u}_d$ selected in each converter, the matrix is expressed as follows:

$$
D_1 = \cdots = D_n = \begin{bmatrix} \frac{2(n-1)KU_{in}K_dK_P}{nL_f} - \frac{2KR_d'}{L_f} & -\frac{2KU_{in}}{L_f} & \frac{2KU_{in}}{L_f} \\ -\frac{(n-1)K_dK_I}{n} & 0 & 0 \\ -\frac{2(n-1)K_dK_P}{0.75nT_s} & \frac{2}{0.75T_s} & -\frac{1}{0.75T_s} \end{bmatrix} \tag{26}
$$

For the relationship between the three state variables $\hat{i}_L$, $\hat{u}_{PI}$, and $\hat{u}_d$ selected in each converter and the state variables selected in other converters, the matrix is expressed as follows:

$$
E_1 = \cdots = E_n = \begin{bmatrix} -\frac{2KU_{in}K_dK_P}{nL_f} & 0 & 0 \\ \frac{K_dK_I}{n} & 0 & 0 \\ \frac{2K_dK_P}{0.75nT_s} & 0 & 0 \end{bmatrix} \tag{27}
$$

For the relationship between the three state variables $\hat{i}_L$, $\hat{u}_{PI}$, and $\hat{u}_d$ selected in each converter and the output capacitor voltage disturbance, the matrix is expressed as follows:

$$
F_1 = \cdots = F_n = \begin{bmatrix} \frac{2KU_{in}K_P}{L_f} + \frac{2KU_{in}K_dK_P}{nR_oL_f} - \frac{1}{L_f} \\ -\frac{K_dK_I}{nR_o} - K_I \\ -\frac{2K_dK_P}{0.75nT_sR_o} - \frac{2K_P}{0.75T_s} \end{bmatrix} \tag{28}
$$

For the n-module paralleled IPOS phase-shifted full-bridge converters system, the relationship between the voltage disturbance $\hat{u}_c{}'$ of the coupled parallel equivalent capacitor and other state variables can be expressed as follows:

$$\hat{u}_c{}' = \frac{1}{snC_f}(\hat{i}_{L1} + \hat{i}_{L2} + \cdots + \hat{i}_{Ln} - \frac{\hat{u}_c{}'}{R_o}) \tag{29}$$

According to (16), the matrix can be set as follows:

$$G_1 = \cdots = G_n = \begin{bmatrix} \frac{1}{nC_f} & 0 & 0 \end{bmatrix} \tag{30}$$

Therefore, the state matrix $A_n$ in the state equation $\dot{x} = A_n x + B_n u$ of the n-module paralleled IPOS phase-shifted full-bridge converters system can be expressed as follows:

$$A_n = \begin{bmatrix} D_1 & E & \cdots & E & F_1 \\ E & D_2 & \cdots & E & F_2 \\ \vdots & \vdots & \ddots & \vdots & \vdots \\ E & E & \cdots & D_n & F_n \\ G_1 & G_2 & \cdots & G_n & -\frac{1}{nR_oC_f} \end{bmatrix} \tag{31}$$

For the input matrix $B_n$, set the matrix as follows:

$$H_1 = \cdots = H_n = \begin{bmatrix} -\frac{2KU_{in}K_P}{L_f} & K_I & \frac{2K_P}{0.75T_s} \end{bmatrix}^T \tag{32}$$

Thus, the input matrix $B_n$ can be expressed as follows:

$$B_n = \begin{bmatrix} H_1 & H_2 & \cdots & H_n & 0 \end{bmatrix}^T \tag{33}$$

## 3. Stability Analysis of the Paralleled IPOS Phase-Shifted Full-Bridge Converters System

Eigenvalue analysis is a method to analyze the stability of the system by solving the eigenvalues of the state matrix [23]. This method can analyze the system stability under many working conditions and find out the parameters related to stability in the system, but the disadvantage is that the calculation of a high-order matrix is more complicated. This paper involves a small matrix order, so this paper chooses to use the eigenvalue analysis method to analyze the stability of the IPOS phase-shifted full-bridge converter parallel system.

The parameters of each converter module in the parallel system are the same, so the parameters can be designed based on the single-module IPOS phase-shifted full-bridge converter. Among them, for the droop coefficient, assuming that the voltage accuracy of the parallel system is 5% because the output voltage is 2000 V, the maximum difference between the actual output voltage and the rated output voltage is 100 V, and the output current is 50 A under full load conditions, so the droop coefficient $K_d$ takes 2. For the selection of PI parameters of the voltage loop, according to Figure 3, the open-loop transfer function $G_{ud}$ from the primary duty cycle $d$ to the output capacitor voltage $u_c$ can be expressed as follows:

$$G_{ud} = \frac{\hat{u}_c}{\hat{d}} = \frac{nKU_{in}R_o}{L_fC_fR_os^2 + (nKR_dR_oC_f + L_f)s + nKR_d + R_o} \tag{34}$$

Then, the open-loop transfer function $G_u$ of the system is expressed as follows:

$$G_u = G_{PI} \cdot e^{-1.5T_s s} \cdot G_{ud} \tag{35}$$

$G_{PI}$ in the above equation is the transfer function of the PI controller. Considering the dynamic performance and steady-state performance of the voltage loop, the final choice

is $K_P = 0.0001$ and $K_I = 0.3$. When $K_P = 0.0001$ and $K_I = 0.3$, the bandwidth in the voltage loop is 140 Hz, the phase margin is 83 deg, and the gain margin is 16.9 dB. The Bode diagram of the voltage loop is shown in Figure 5.

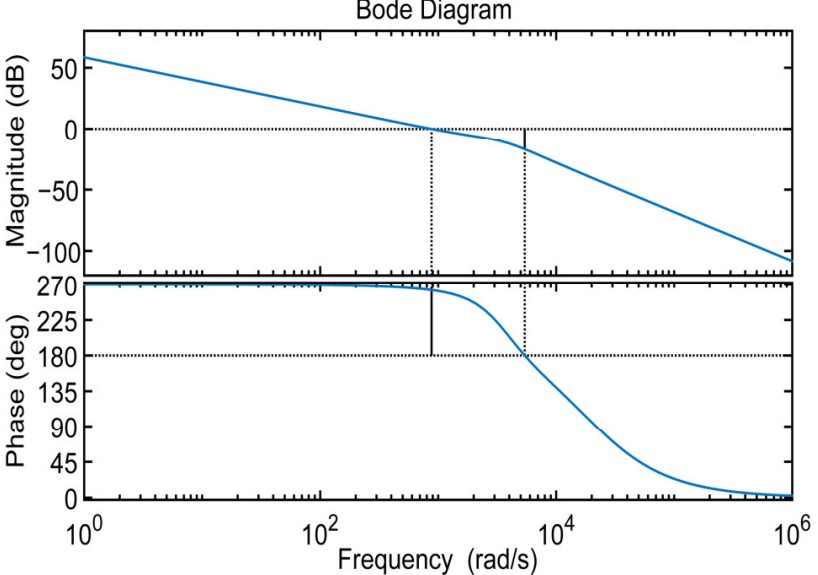

**Figure 5.** Bode diagram of the voltage loop for $K_P = 0.0001$ and $K_I = 0.3$.

From the state matrix $A_1$, the droop coefficient and the controller parameter change will affect the position of the characteristic root, which, in turn, affects the system stability. However, many papers have studied this phenomenon. Therefore, this paper focuses on the influence of load and the number of parallel converters on the stability of the system.

The module parameters of the single-module IPOS phase-shifted full-bridge converter system are shown in Table 1.

**Table 1.** Parameters of single IPOS phase-shifted full-bridge converter.

| Parameter Name | Sign | Value |
| --- | --- | --- |
| Input voltage | $U_{in}$ | 240 V |
| Output voltage | $U_o$ | 2000 V |
| Ratio of transformer | 1:$K$ | 1:6 |
| Switching frequency | $f_s$ | 15 kHz |
| Sampling period | $T_s$ | 66.67 μs |
| Transformer leakage inductance | $L_{lk}$ | 0.3 μH |
| Switching tube parasitic capacitance | $C_r$ | 3 nF |
| Output filtering inductance | $L_f$ | 274 μH |
| Output filter capacitor | $C_f$ | 35 μF |
| Rated power | $P_o$ | 100 kW |
| Droop coefficient | $K_d$ | 2 |
| Proportional coefficient | $K_P$ | 0.0001 |
| Integral coefficient | $K_I$ | 0.3 |

### 3.1. The Influence of Load Change on System Stability

Taking the two-module paralleled IPOS phase-shifted full-bridge converters system as the object, the influence of load power change on the stability of the system is first studied. The load power $P_o$ is reduced from 100 kW to 1 kW, and the eigenvalue trajectory is shown in Figure 6.

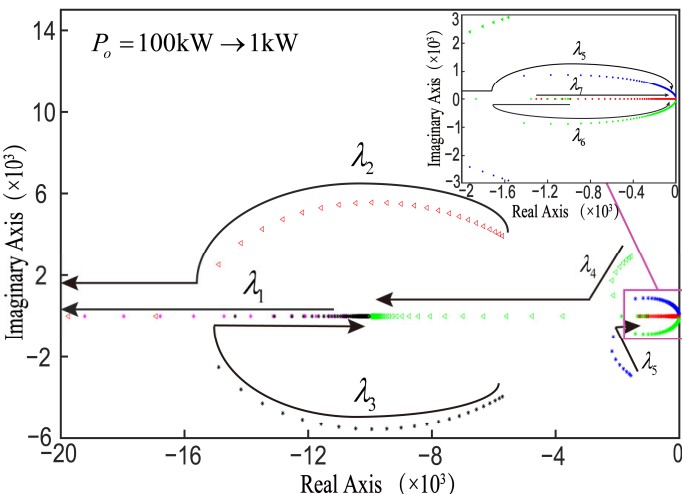

**Figure 6.** Trajectory diagram of characteristic root changing with load.

As shown in Figure 6, there are seven characteristic roots in the system. As the load power decreases, the four characteristic roots of $\lambda_1 \sim \lambda_4$ are always far from the imaginary axis. Therefore, the changes in the three dominant characteristic roots of $\lambda_5$, $\lambda_6$, and $\lambda_7$ are mainly concerned. $\lambda_5$, $\lambda_6$, and $\lambda_7$ are constantly approaching the imaginary axis, indicating that the decrease in load power will reduce the stability margin of the system, i.e., the parallel system using current-sharing control will have poor stability and an obvious decline in dynamic performance under light load conditions.

### 3.2. The Influence of the Number of Paralleled IPOS Phase-Shifted Full-Bridge Converters on Stability

With the further increase in the power supply scale, the DC converter needs to transmit higher and higher power levels, and the number of converters that need to be connected in parallel also needs to be increased. When the number of converters in parallel is further increased, whether the system has stability problems also needs to be further explored.

According to (31), the state matrices $A_1$ and $A_2$ of the single-module IPOS phase-shifted full-bridge converter system and the two-module paralleled IPOS phase-shifted full-bridge converters system are obtained, respectively. The load power $P_o$ is gradually reduced from 100 kW to 1 kW, and the dominant eigenvalues of the two are analyzed. The trajectory is shown in Figure 7.

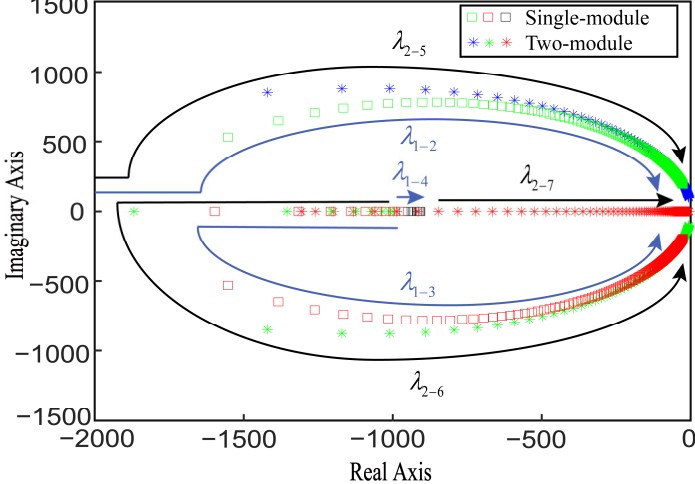

**Figure 7.** The characteristic roots of the single-module and two-module paralleled converters system change with Po.

In Figure 7, $\lambda_{2-5}$, $\lambda_{2-6}$, and $\lambda_{2-7}$ are the dominant eigenvalues of the two-module paralleled IPOS phase-shifted full-bridge converters system. For the single-module IPOS phase-shifted full-bridge converter system, it has four eigenvalues. As the load power decreases, the three eigenvalues of $\lambda_{1-2}$, $\lambda_{1-3}$, and $\lambda_{1-4}$ are close to the imaginary axis, which has a great influence on the system and is the dominant eigenvalue of the system. When the load power $P_o$ gradually decreases from 100 kW to 1 kW, compared with the single-module IPOS phase-shifted full-bridge converter system, the conjugate eigenvalues $\lambda_{2-5}$ and $\lambda_{2-6}$ of the two-module paralleled IPOS phase-shifted full-bridge converters system are closer to the imaginary axis than $\lambda_{1-2}$ and $\lambda_{1-3}$. The speed of $\lambda_{2-7}$ on the real axis is also much larger than the speed of $\lambda_{1-4}$ close to the imaginary axis.

Therefore, it can be concluded that the number of parallel connections of IPOS phase-shifted full-bridge converters has a certain influence on the stability, and the increase in the number of parallel connections will reduce the stability of the system.

## 4. Control Strategy Optimization of the Paralleled IPOS Phase-Shifted Full-Bridge Converters System Based on the PSO Algorithm

The above analysis proves that when the droop method is used for current-sharing control, the stability of the system is poor under light load conditions, and the increase in the number of converters in parallel will further reduce the stability of the system. For the eight-module paralleled IPOS phase-shifted full-bridge converters system, when the load power $P_o$ is 1 kW, $K_d = 2$, $K_P = 0.0001$, and $K_I = 0.3$, the system damping under light load is small, the output voltage oscillation attenuation is slow, and the system stability is poor.

Therefore, it is necessary to optimize the eight-module paralleled IPOS phase-shifted full-bridge converters system to improve its stability under a light load. In engineering, the method of paralleling dead load $R_{od}$ on the output side is generally used to avoid the system from entering the working state of too small a load, but the selection of $R_{od}$ needs to consider many factors. When the resistance of $R_{od}$ is too small, the load will greatly increase the power consumption of the system and reduce the operating efficiency of the system. When the resistance of $R_{od}$ is too large, it cannot ensure that the system has excellent stability and dynamic performance.

In this paper, the PSO algorithm is used to optimize the control parameters so that the dominant pole of the system is far away from the imaginary axis and the damping ratio is as large as possible to improve the stability of the system and replace the parallel dead load on the output side.

### 4.1. Establish the Objective Function

According to (31), the state space matrix $A_8$ of the eight-module paralleled IPOS phase-shifted full-bridge converters system can be determined, which is a $25 \times 25$-order matrix, and the eigenvalues of the system can be further solved. The characteristic root is written as $\lambda_i = a_i + b_i j$, where the real part $a_i$ corresponds to the distance between the characteristic root and the imaginary axis, reflecting the speed of attenuation. The larger the amplitude, the faster the attenuation, and the better the stability of the system. The damping ratio $\zeta_i = -a_i / \sqrt{a_i^2 + b_i^2}$ reflects the attenuation rate of the oscillation amplitude. The larger the damping ratio, the faster the oscillation attenuation. Therefore, the optimization objective of the system control parameters can be set as follows: (1) all the eigenvalues of the system are in the left half plane and far from the imaginary axis; (2) the damping ratio of complex eigenvalues is improved. Therefore, the optimization objective function of the eight-module paralleled IPOS phase-shifted full-bridge converters system can be expressed as follows:

$$\min F = \sum_{i=1}^{N} [f_a(\lambda_i) + f_b(\lambda_i)] \tag{36}$$

Among them, $N$ is the number of characteristic roots, $f_a(\lambda_i)$ and $f_b(\lambda_i)$ are expressed as follows:

$$f_a(\lambda_i) = \begin{cases} 0 & a_i < a_d \\ \gamma_i(a_i - a_d) & \text{others} \end{cases} \tag{37}$$

$$f_b(\lambda_i) = \begin{cases} 0 & \zeta_i > \zeta_d \\ \beta_i(\zeta_d - \zeta_i) & \text{others} \end{cases} \tag{38}$$

In the formulas, $a_d$ represents the real part of the optimization target of the characteristic root, $\zeta_d$ represents the optimization target damping ratio of the characteristic root, $\gamma_i$ and $\beta_i$ represent the weights of the optimization target one and the optimization target two, respectively, and the corresponding coefficients are changed according to the distance between the characteristic root and the imaginary axis and the damping ratio to ensure that the objective function approaches the optimal value. The specific performance is that the closer the characteristic root is to the imaginary axis, the larger the $\gamma_i$ is, the smaller the damping ratio is, and the larger the $\beta_i$ is.

*4.2. Optimize Control Parameters*

According to (36), combined with the PSO algorithm, the control parameters of the eight-module paralleled IPOS phase-shifted full-bridge converters system are optimized. For the PSO algorithm, the position of the particle represents the parameters of the controller, the speed of the particle represents the adjustment of the controller parameters, and the fitness value of the particle represents the objective function value. In this algorithm, the position of the particle corresponding to the global extremum is the final optimal control parameter value. The block diagram of the optimized controller is shown in Figure 8, and the specific algorithm flow is shown in Figure 9.

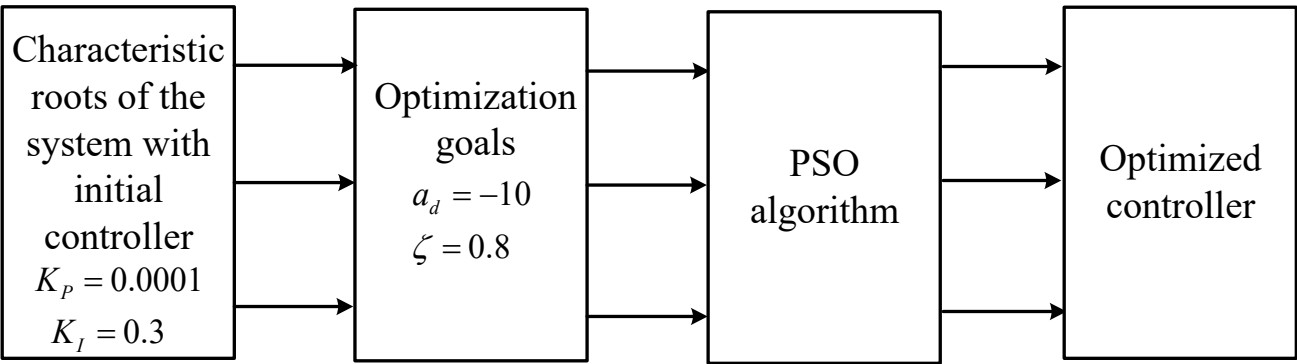

**Figure 8.** Block diagram of the optimized controller.

Table 2 shows the optimization algorithm parameters such as the space dimension $D$, the maximum number of iterations $k_{\max}$, the number of particles $M$, inertia weight $\omega$, acceleration factor $c_1$ and $c_2$, the optimization target real part $a_d$ of the characteristic root, and the optimization target damping ratio $\zeta_d$ of the characteristic root, as well as the corresponding weight $\gamma_i$ at different $a_i$ and the corresponding weight $\beta_i$ at different $\zeta_i$.

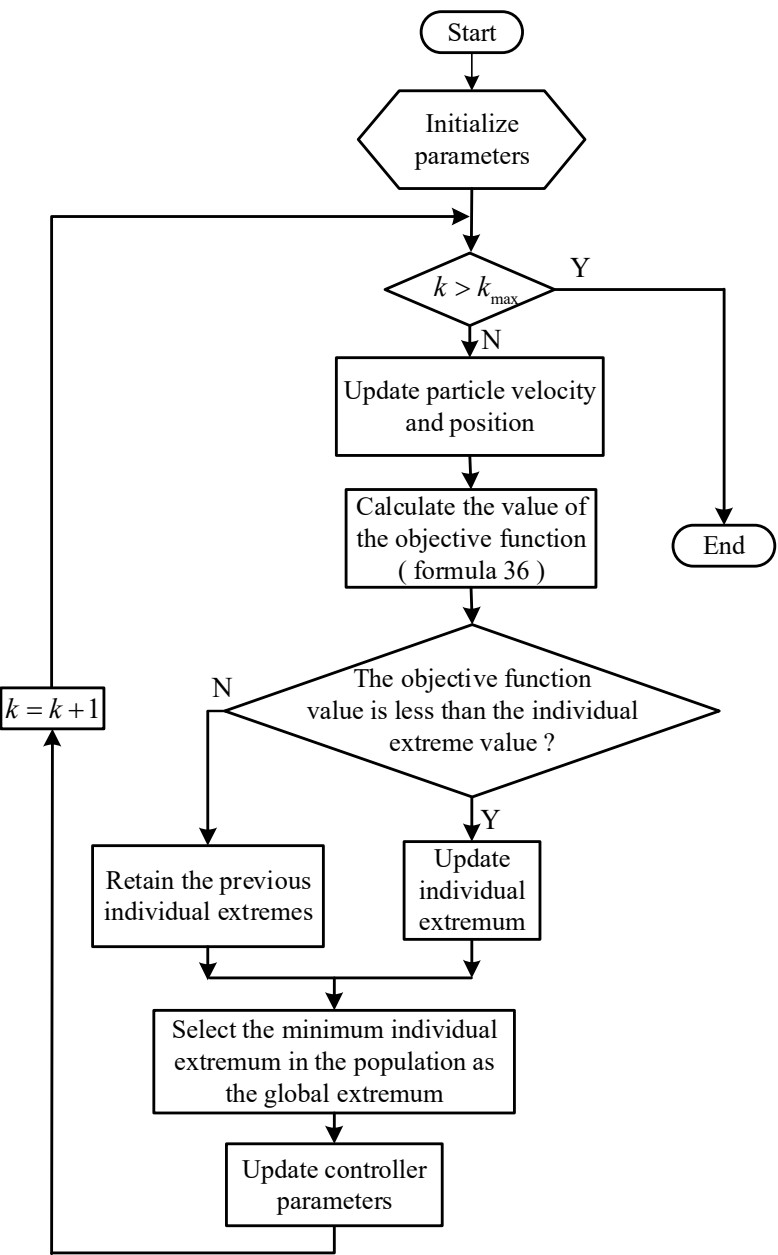

**Figure 9.** Flow chart of PSO algorithm.

**Table 2.** The parameters of optimization algorithm.

| Parameter | Value | Parameter | Value |
|---|---|---|---|
| $D$ | 2 | $k_{\max}$ | 100 |
| $M$ | 20 | $\omega$ | 1 |
| $c_1$ | 2 | $c_2$ | 2 |
| $a_d$ | $-10$ | $\zeta_d$ | 0.8 |
| $\gamma_i$ | $3, a_i \in [-3, 0)$<br>$2, a_i \in [-7, -3)$<br>$1, a_i \in [-12, -7)$ | $\beta_i$ | $3, \zeta_i \in (0, 0.2]$<br>$2, \zeta_i \in (0.2, 0.5]$<br>$1, \zeta_i \in (0.5, 0.8]$ |

Based on the above optimization algorithm, the optimized controller parameters and the initial controller parameters are calculated, as shown in Table 3.

**Table 3.** Optimization results of control parameters.

| Controller Parameter | Initial Parameters | Optimized Parameters |
|---|---|---|
| Proportional coefficient $K_P$ | 0.0001 | 0.038 |
| Integral coefficient $K_I$ | 0.03 | 9.71 |

The dominant eigenvalue and damping ratio corresponding to the control parameters before and after the optimization of the system are calculated, respectively, as shown in Table 4.

**Table 4.** Dominant eigenvalues and damping ratios before and after optimization of control parameters.

| Dominant Characteristic Root | Pre-Optimization Value | Optimized Value | Damping Ratio before and after Optimization |
|---|---|---|---|
| $\lambda_1, \lambda_2$ | $-2.79 \pm 68.14j$ | $-307.8 \pm 256.06j$ | $0.04 \to 0.768$ |
| $\lambda_3 \sim \lambda_9$ | $-0.33$ | $-10.13$ | / |

As shown in Table 4, the dominant characteristic root optimized via the PSO algorithm is farther from the imaginary axis than before optimization, and the damping ratio increases greatly. The real part and damping ratio of the dominant characteristic root after optimization are close to the optimization target, which shows that the optimization effect is good.

## 5. Simulation Verification

The simulation model of the n-module paralleled IPOS phase-shifted full-bridge converters system is built in a detailed MATLAB/Simulink simulator, shown in Figure 10, and the parameters are set according to Table 1.

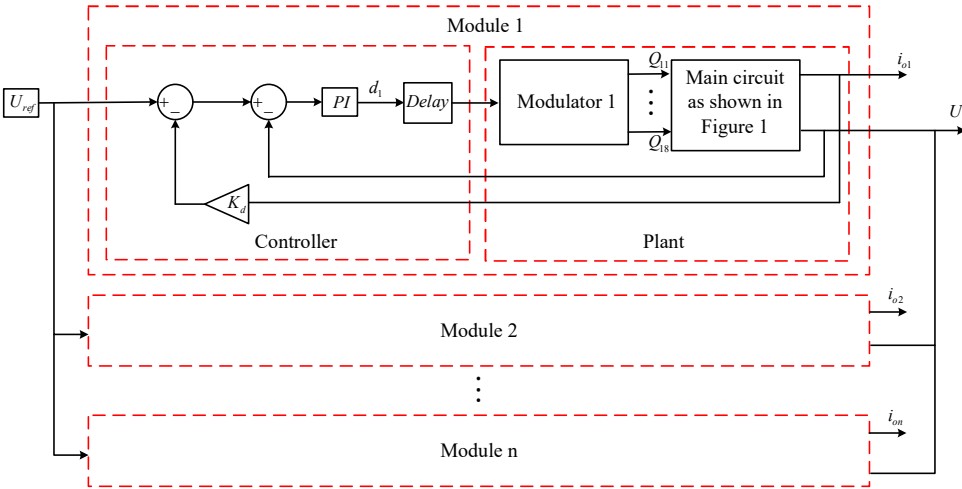

**Figure 10.** MATLAB/Simulink simulation model of the n-module paralleled IPOS phase-shifted full-bridge converters system.

In the above figure, $U_{ref}$ is the reference voltage of the voltage loop, which is 2000 V, and the Modulator realizes the PWM switching modulation of the switching tubes of the module.

To verify the influence of the load on the stability of the system, the simulation model of the two-module paralleled IPOS phase-shifted full-bridge converters system is first built. $K_P = 0.0001$, $K_I = 0.3$ and $K_d = 2$ are set up. Simulations are performed with a load power $P_o$ of 4 kW, 2 kW, and 1 kW, respectively, and a 4 kW load is put into the simulation at 0.7 s, lasting 2 ms, which is equivalent to a small disturbance. The simulation results are shown

in Figure 11. The simulation results shows that the lighter the load, the greater the impact of the disturbance on the system, and the longer the time required to restore stability after the disturbance. The conclusion that the decrease in the load will make the stability of the parallel system worse is verified.

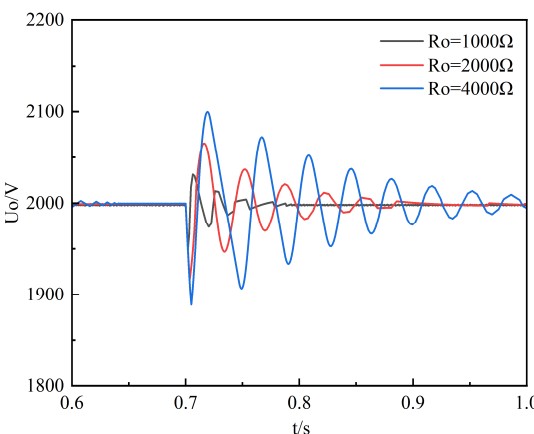

**Figure 11.** Small disturbance simulation waveforms of different $P_o$.

Figure 12 shows the output voltage simulation waveforms of the two-module paralleled converters system, the three-module paralleled converters system, and the eight-module paralleled converters system when the load power $P_o$ is 1 kW, $K_P = 0.0001$, $K_I = 0.3$, and $K_d = 2$. It is found that the waveform presents a sawtooth waveform, which is due to the high overshoot of the output voltage during the start-up phase. When the output voltage is higher than the rated voltage, the filter capacitor discharges. However, due to the slow energy consumption of the load under light load conditions, the output voltage drop edge is relatively flat, and the overall waveform is biased toward the sawtooth wave.

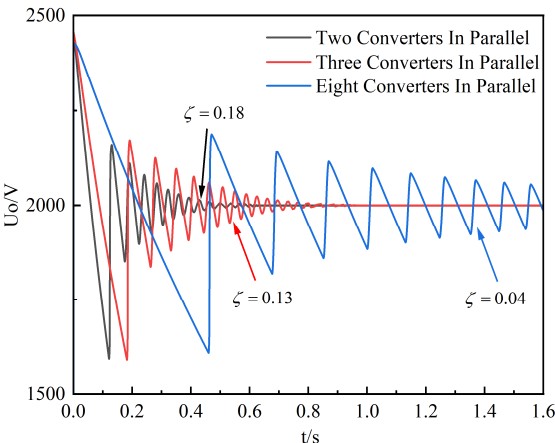

**Figure 12.** Output voltage simulation waveforms of paralleled systems composed of different numbers of converters.

With the increase in the number of converters in parallel, the damping ratio of the system gradually decreases, the oscillation frequency of the system obviously decreases, the oscillation attenuation of the output voltage slows down, and the stability of the system deteriorates.

For the optimization of the control parameters, Figure 13 shows the output voltage simulation waveforms of the eight-module paralleled IPOS phase-shifted full-bridge converters system with the control parameters before and after optimization under the load power $P_o$ is 1 kW and $K_d = 2$, respectively. Compared with the output voltage waveform before optimization, the optimized output voltage oscillation attenuation is significantly accelerated, the damping ratio is significantly increased, and the system stability is significantly

improved. The proposed control parameter optimization method of the eight-module paralleled IPOS phase-shifted full-bridge converters system has a significant effect, which verifies the feasibility of improving the stability of the system by optimizing the parameters instead of the output side parallel dead load under light load.

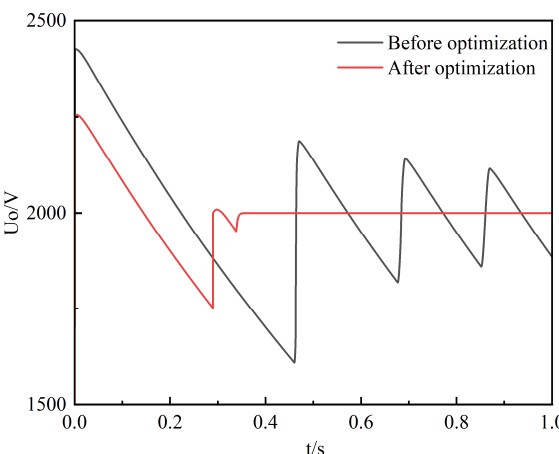

**Figure 13.** Simulation waveforms of output voltage before and after optimization of eight-module paralleled IPOS phase-shifted full-bridge converters system.

## 6. Conclusions

This paper presents the theoretical stability analysis and control strategy optimization of the paralleled IPOS phase-shifted full-bridge converters system based on droop control. According to the theoretical results and simulation, the stability of the paralleled IPOS phase-shifted full-bridge converters system will be greatly reduced under light load conditions, and when the number of converters in parallel is further increased, the stability of the system will be correspondingly worse. Focusing on this phenomenon, an optimal control strategy based on the PSO algorithm is proposed, which can significantly improve the stability of the parallel system of multiple IPOS phase-shifted full-bridge converters under light load conditions. This method can replace the dead load on the output side of the paralleled system under the light load condition to reduce power consumption, which has guiding significance for actual engineering.

**Author Contributions:** Z.Q. constructed the model of the whole system, studies its stability, and optimizes it; Z.Q. carried out the research and analyzed the numerical data using guidance from H.C. and X.L.; Z.Q., H.C. and X.L. collaborated to prepare the manuscript. All authors have read and agreed to the published version of the manuscript.

**Funding:** This research was funded by the National Natural Science Foundation of China under Grant 61473116.

**Data Availability Statement:** The data presented in this study are available on request from the corresponding author.

**Conflicts of Interest:** The authors declare no conflict of interest.

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
