# Peer review of "Stability Analysis and Control Strategy Optimization of a Paralleled IPOS Phase-Shifted Full-Bridge Converters System Based on Droop Control"

_electronics, doi:10.3390/electronics12173685_

Round 1

Reviewer 1 Report

This is a great research topic. The idea presented in the paper is very interesting, however significant changes must be made in this article.

1- The wording in the abstract should change and pay attention to punctuation. 

2- The introduction is too short, and there are not enough supporting materials. 

3- How did the authors get to figure 3? Can you show your derivation?

4- Please show the derivation, at least the main equation that was used to obtain equation (1)

5- What is Kd in figure 4?

6- A final block diagram of the optimized controller should be added with all the appropriate signals.

7- In the abstract the authors talk about the current sharing issue, however in simulation you only show voltage stability issues. Can the results about the current sharing improvement be added also?

8- Please rewrite the conclusion. 

9- Can all the claims be supported by experimental results not just simulation? If yes, can the experimental results: picture of the set or HIL and scope results be shown? 

Please pay attention on punctuation, choice of words and run on sentences. 

Reviewer 2 Report

The submitted manuscript presents the stability analysis of paralleled input parallel output series phase-shifted full-bridge converters system based on droop control. The topic is rather interesting both for the power electronics and the control communities. Overall, the manuscript is well written and it clearly presents the objective of the Authors' research. However, several comments are due.

In the Introduction Authors could further improve their literature review by recalling further research dealing with control of DC/DC power converters. For instance, in [Supercapacitor stability and control for More Electric Aircraft application, 2020 European Control Conference (ECC), St. Petersburg, Russia, 2020, pp. 1909-1914] and [Stability and Control for Buck–Boost Converter for Aeronautic Power Management. Energies 2023, 16, 988] stability analysis and control design are discussed for bidirectional DC/DC converter for aeronautic applications.

In Section 2.1, Authors directly present the small-signal model of the IPOS phase-shifted full-bridge converter. However, the small-signal model is usually obtained starting from the system of differential equations describing the dynamics of the converter. In fact, as it is presented in the current version of the manuscript, it is not clear how the small-signal model is retrieved and, hence, it is not clear how it can be guaranteed that the presented small-signal model is actually descriptive of the converter behavior. Authors should provide the system of (perhaps nonlinear) differential equations describing the IPOS phase-shifted full-bridge converter first and then construct the small-signal system starting from it.

Does equation (2) come from the Padé approximation of models with time delay? Please be more specific and add a reference to let the reader further understand your reasoning

In (18), matrix An is presented. However, in order to fully describe the linearized state-space system, also Bn should be presented.

Line 203, I believe Authors meant "imaginary axis" in place of "unit circle".

In the preliminary simulation, the PI gains were chosen as Kp=0.0001 and Ki=0.3. How were these parameters chosen? Just trial and error? Moreover, for the sake of understanding the simulation results, matrix A1 should be presented in the section.

In Section 3.1, the influnce of the load change on system stability is discussed. While the load is presented in terms of resistive load Ro in Section 2, in Section 3.1 it seems that Authors are considering constant power loads. Authors should better describe whether they consider resistive loads or constant power loads.

Section 4.1 could be entirely removed as Particle Swarm Optimization algorithm can be assumed to be of common knowledge for the readers of the Electronics journal.

It is not clear why, while the stability analysis was performed for a single converter, 8 converters are instead considered to establish the objective function.

For the Simulation in Section 5, which simulation software was used? If it was Simulink, did the Authors use some specific toolbox (such as SimPowerSystems or Simscape) or they simply implemented the small gain system equations in the simulation environment? Authors should better clarify how the simulation results were obtained.

It would be very interesting to show some plots of the phase shift signals in the case of tuned and non-tuned control gains.

Reviewer 3 Report

The topic in which this paper focuses is very interesting and emerging, in terms of Power Electronics applications. The manuscript is of good quality in general, well-organized and well-writen. Although, I think that there are some minor points that have to be highlighted and better elaborated, in order to improve the paper quality. My comments are the following:

(1) Define all symbols and parameters the first time you use them, in the modeling and mathematical analysis procedure. It will be more helpful for the reader. 

(2) What do you mean by the phrase "In the case of no-load, three resistors Ro are connected to 1000Ω, 2000Ω and 4000Ω respectively, that is, the load power Po is 4kW, 2kW and 1kW respectively, and a 4kW load is put into the simulation at 0.7s, lasting 2ms, which is equivalent to a small disturbance."? What do you mean by the term "no load" and why three resistors are connected? Please describe in more detail. Also, a figure including the simulation model would be helpful for the reader to understand. 

(3) What is the "Julie criterion", stated in line 200 and how is it utilized, to obtain the PI coefficients? Please elaborate in detail and use additional appropriate references. Also, a Bode plot would be helpful, in order to provide the initial (before the PSO algorithm) controller bandwidth, gain margin and phase margin. 

(4) Why is the sampling frequency selected to be equal to the switching frequency? Shouldn't it be higher enough than the switching frequency? Does it affect the stability results? Please elaborate. 

1) The language level is of good quality in general. However, there are several minor grammar, typo and syntax mistakes in the manuscript. Please carefully proofread your paper in order to correct all these mistakes.

2) In Figure 1, correct "moudle" to "module" and number each PSFB module accordingly.

Round 2

Reviewer 1 Report

Great improvement! 

1- Figure 10 should be expanded by showing all the intermediate control signals.

Great Improvement!

Reviewer 2 Report

Authors have correctly replied to the questions from the previous review round. No further changes are required.

Author Response

Thanks again for your professional review work on our article.